# Phenotypes associated with genes encoding drug targets are predictive of clinical trial side effects

Phuong A. Nguyen[1], David A. Born[1], Aimee M. Deaton [1], Paul Nioi[1,2] & Lucas D. Ward [1,2]

Only a small fraction of early drug programs progress to the market, due to safety and efficacy failures, despite extensive efforts to predict safety. Characterizing the effect of natural variation in the genes encoding drug targets should present a powerful approach to predict side effects arising from drugging particular proteins. In this retrospective analysis, we report a correlation between the organ systems affected by genetic variation in drug targets and the organ systems in which side effects are observed. Across 1819 drugs and 21 phenotype categories analyzed, drug side effects are more likely to occur in organ systems where there is genetic evidence of a link between the drug target and a phenotype involving that organ system, compared to when there is no such genetic evidence (30.0 vs 19.2%; OR = 1.80). This result suggests that human genetic data should be used to predict safety issues associated with drug targets.

[1] Amgen, Inc., 360 Binney St., Cambridge, MA 02142, USA. [2]Present address: Alnylam Pharmaceuticals, Inc., 300 Third St., Cambridge, MA 02142, USA. These authors contributed equally: Phuong A. Nguyen, David A. Born. Correspondence and requests for materials should be addressed to L.D.W. (email: lward@alnylam.com)

Approximately 90% of drug candidates fail to progress through clinical trials because of issues with safety or efficacy[1–4] and this problem is magnified for novel targets[5–7]. Safety is an especially big hurdle for drug development because molecules' safety must be tested in preclinical species before reaching humans, and an unacceptable side-effect profile may not become apparent until clinical testing because of the poor translatability from animal studies[8,9]. As a result, it has been estimated that although the median cost of developing a new drug from Phase I to approval is around $250 million, the total R&D cost per new drug is over $2.5 billion[10]. Therefore it is critical to develop better preclinical assessments for choosing only the safest drug targets and ideal molecules for clinical testing, and even modest improvements in preclinical predictions of toxicity can be massively valuable to drug development success[9].

There are many factors that contribute towards the safety and efficacy profile of a drug, and many of these have been studied through retrospective analyses to improve drug design. These factors include chemical properties of the drug that affect its pharmacokinetics and pharmacodynamics, biological properties of the target such as its expression pattern, the activity profile of the drug against off-target proteins, and genetic differences between patients that modulate inter-individual differences in response[11–23]. While all of these factors can contribute to the safety profile of a drug, the physiological role of the intended target is the most obvious and unavoidable consideration. While some drug targets have very narrow and specific roles in certain diseases, others can have broader biological roles—meaning that when they are drugged, there can be numerous unintended effects. One way of understanding biological roles of the target has been though human genetics, with the idea that phenotypes arising from natural variation in the gene encoding a protein will predict the phenotypes that would result from drugging that protein. Family studies of rare Mendelian syndromes or genome-wide association studies (GWAS) of common diseases can both serve as natural experiments that point to new targets to pursue (with PCSK9 being a celebrated example)[4,24–26]. Retrospective analysis has shown that this genetic validation method results in finding efficacious targets and indications that are more likely to succeed through development and reach patients[27]. However, no such retrospective analysis has been published linking the genetic phenotypes of drug targets—which may include additional, pleiotropic phenotypes besides the therapeutic indication—to corresponding unintended side effects.

In this study to explore this relationship between drug targets' genetic associations and drugs' side effects, we compile a set of drugs for which data are available for both (a) the drugs' intended human target proteins and (b) the side effects observed during clinical trials of that drug. We then compare those side effects to the phenotypes associated with mutations in the genes encoding those target proteins. Thus, we investigate the ability of human genetics to predict the adverse events and side effects that will arise when drugging given target proteins.

## Results

**Enrichment analysis**. We merged data from several databases to compare genetic phenotypes with side effects (Fig. 1). To examine the ability of genetics to predict drug treatment-related side effects we first needed a large set of standardized side effect information. We obtained drug, indication, and side effect information from the Cortellis Clinical API and Cortellis Drug Design API (Clarivate Analytics, Inc.) for 31,194 clinical trials, all of which had at least one clinical side effect recorded in a controlled vocabulary of phenotype terms. We excluded oncology trials because of cancer drugs' inherent cytotoxicity and

difference in acceptable side effect profiles[28], we excluded combination therapies to reduce complexity in our analysis, and we limited our analysis to small molecule and biologic drugs. We also excluded very commonly-observed side effects such as headache (see Methods) to increase the sensitivity of our analysis. We then identified the accepted human protein targets of these drugs using the union of three databases: Drugbank[29–33], Pharmaprojects (Informa PLC), and a recently published curated map of drugs' molecular efficacy targets[33]. A total of 1819 of the drugs had at least one target annotated by one of these databases, and there were 1046 unique targets pursued. For the genes encoding these drug targets, we obtained associated phenotypes from two sources: information from genome-wide association studies (GWAS) collated by the STOPGAP database[34], which uses coding and noncoding variant annotations to assign target genes to GWAS SNPs; and Mendelian information collated by the Human Phenotype Ontology (HPO)[35], specifically the subset of HPO phenotypes from Online Mendelian Inheritance in Man (OMIM)[36] which is mapped to controlled phenotype terms. A total of 1394 of the 1819 drugs had at least one target with genetic phenotype information; 641 of 1046 unique targets pursued had genetic phenotype information. All of the 1819 drugs and 1046 targets were retained in our analysis, in order to study the correlates of the presence or absence of genetic information for a target.

In order to examine whether the genetic phenotypes of these drug targets matched the drug's side effects for each drug, we mapped its indication(s), side effect(s), and targets' genetic associations to 21 phenotype categories using the MedDRA ontology at the system organ class (SOC) level. We classified each of these drug-organ system pairs as (a) whether the drug has ever been pursued for a therapeutic indication involving that organ system, (b) whether that drug has ever elicited a clinical side effect involving that organ system, and (c) whether that drug's targets have genetic phenotype associations involving that organ system (Fig. 1; see Supplementary Data 1 for full dataset).

We found a striking relationship between drug target genetics and the organ systems in which side effects were observed (Fig. 2, Supplementary Table 1, Supplementary Table 2). Across all 38,199 drug–side effect combinations, the existence of a genetic association between a drug's target(s) and phenotypes in a given organ system increases the probability that a side effect will be observed in that organ system during clinical trials from 19.2 to 30.0% (OR = 1.80; 95% CI = 1.71–1.90, Fisher's exact $P = 1.7 \times 10^{-94}$). This association could be largely driven by the correspondence between genetics and drug indications; side effects may tend to occur or be monitored more closely in the organ system of drug action, or side effects may be exaggerated pharmacology effects directly related to the therapeutic effect. To exclude this confounding effect, we performed the same enrichment test while removing from our analysis 4710 drug-organ system pairs, where the organ is the intended site of therapeutic efficacy for one of the drug's pursued indications, and found that genetic associations increase the probability of a side effect in the corresponding off-indication organ system from 15.6 to 22.4% (OR = 1.55; 95% CI = 1.45–1.66; Fisher's exact $P = 1.70 \times 10^{-36}$). This decreased off-indication effect confirms that some, but not all, of the genetics–side effect relationship is driven either by aspects of the intended therapeutic biology of the drug, or by other effects in the same organ system. To investigate potential biases in the composition of our dataset we performed separate permutation analyses of the target genetics and drug side effects (Supplementary Table 2; Supplementary Figure 5). In each analysis, there was indeed residual overlap of genetic and pharmacological phenotypes; however, neither the randomization of genetics nor the randomization of side effects

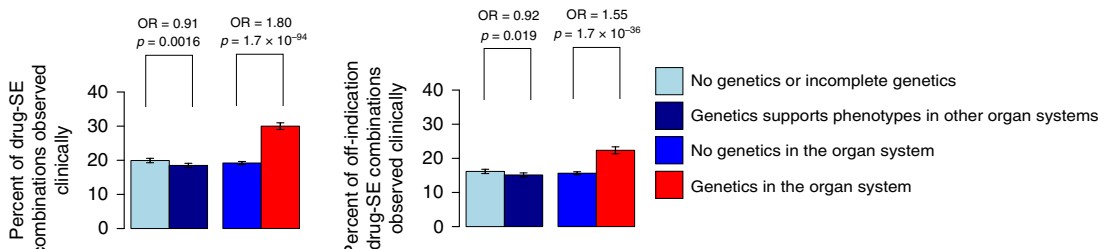

**Fig. 1** Data sources and analysis workflow

**Fig. 2** Rate of side effect (SE) manifestation across drug-SE combinations classified by genetic support. OR, odds ratio; P, P-value from Fisher's exact test (two-tailed). The left panel shows analyses based on all 38,199 drug-SE combinations; the right panel shows analyses based on 33,489 "off-indication" drug-SE combinations. Error bars represent the 95% confidence interval of the reported proportions. Source data and confidence intervals for the OR values are shown in Supplementary Table 1. Similar analyses for Mendelian and GWAS genetics separately are shown in Supplementary Figure 4

approached the effect size seen when including true target genetics and drug side effects, confirming the contribution of genetic information to the enrichment, relative to confounding biases (Monte Carlo empirical $P < 0.001$ for each analysis; Supplementary Table 2 and Supplementary Figure 5). We also found that our results were insensitive to whether each of the drug target databases was used individually rather than being combined (Supplementary Table 2).

**Table 1 Analysis of enrichment between genetic phenotypes and side effects in individual organ systems**

| Phenotype category | Drugs with genetics and side effect | Drugs with genetics and no side effect | Drugs with no genetics and side effect | Drugs with no genetics and no side effect | Rate of side effect when genetic association exists | Rate of side effect when genetic association does not exist | OR | P |
|---|---|---|---|---|---|---|---|---|
| Blood | 96 | 269 | 234 | 1220 | 26.3% | 16.1% | 1.86 | 1.34E−05 |
| Heart | 145 | 176 | 451 | 1047 | 45.2% | 30.1% | 1.91 | 3.92E−07 |
| Congenital | 17 | 572 | 22 | 1208 | 2.9% | 1.8% | 1.63 | 0.165 |
| Ear | 5 | 112 | 74 | 1628 | 4.3% | 4.3% | 0.98 | 1.00 |
| Endocrine | 97 | 252 | 177 | 1293 | 27.8% | 12.0% | 2.81 | 3.71E−12 |
| Eye | 56 | 256 | 193 | 1314 | 17.9% | 12.8% | 1.49 | 0.0186 |
| Gastrointestinal | 155 | 276 | 497 | 891 | 36.0% | 35.8% | 1.01 | 0.954 |
| Hepatobiliary | 20 | 153 | 160 | 1486 | 11.6% | 9.7% | 1.21 | 0.423 |
| Immune | 113 | 335 | 251 | 1120 | 25.2% | 18.3% | 1.51 | 1.76E−03 |
| Infection | 64 | 104 | 536 | 1115 | 38.1% | 32.5% | 1.28 | 0.144 |
| Metabolism | 228 | 400 | 280 | 911 | 36.3% | 23.5% | 1.85 | 1.34E−08 |
| Musculoskeletal | 217 | 435 | 322 | 845 | 33.3% | 27.6% | 1.31 | 0.0118 |
| Neoplasm | 48 | 305 | 122 | 1344 | 13.6% | 8.3% | 1.73 | 3.10E−03 |
| Nervous | 343 | 421 | 401 | 654 | 44.9% | 38.0% | 1.33 | 3.73E−03 |
| Pregnancy | 7 | 268 | 35 | 1509 | 2.5% | 2.3% | 1.13 | 0.827 |
| Mental | 208 | 350 | 255 | 1006 | 37.3% | 20.2% | 2.34 | 5.16E−14 |
| Urologic | 75 | 245 | 220 | 1279 | 23.4% | 14.7% | 1.78 | 2.20E−04 |
| Reproductive | 75 | 199 | 186 | 1359 | 27.4% | 12.0% | 2.75 | 5.75E−10 |
| Respiratory | 140 | 211 | 464 | 1004 | 39.9% | 31.6% | 1.44 | 3.68E−03 |
| Skin | 122 | 224 | 413 | 1060 | 35.3% | 28.0% | 1.40 | 8.75E−03 |
| Vascular | 281 | 299 | 436 | 803 | 48.4% | 35.2% | 1.73 | 8.31E−08 |
| **Total** | **2512** | **5862** | **5729** | **24,096** | **30.0%** | **19.2%** | **1.80** | **1.73E−94** |

P-value is from Fisher's exact test (two-sided). Source data and confidence intervals are provided in Supplementary Table 2
OR odds ratio

Looking at individual organ systems (Table 1 and Supplementary Table 2), we see that all statistically-significant effects are in the direction of enrichment, and that the strength of the enrichment detected is highly variable; for example, while gastrointestinal and cardiac side effects are assigned to a similar number of drugs and genes across our dataset, we find a strong signal of enriched off-indication cardiac side effects with target genetics (OR = 2.18, Fisher's exact $P = 1.7 \times 10^{-94}$) and no significant enrichment for gastrointestinal side effects (OR = 1.12, Fisher's exact $P = 0.37$). These differences may be due to the relative contribution of on- and off-target effects to these organ systems, or to the importance of modality to these organ systems; gastrointestinal side effects, for example, may be more often mediated by the absorption of a drug than its effect on its target.

We then explored whether there was any predictive value in the lack of a genetic association. When the gene encoding a drug target does not correlate with a specific phenotype, this could be for two reasons: either perturbing the gene causes exclusively other phenotypes, or there is no information relating to perturbation of the gene (which could be due to a lack of variation, or due to a lack of detectable effects when the gene is mutated.) We were interested in the difference between these two scenarios: if a drug target gene has been correlated with some phenotypes but not others, should drug developers accept this as evidence that those non-associated phenotypes are indeed less likely to manifest when the gene product is drugged? To test this difference, we compared drug–side effect combinations where there was no genetic support, and split these into two sets: those for which there was some genetic information for all of the targets of the drug and those for which there were targets with no genetic associations and potentially missing information. As expected, a drug's targets having genetic support for other phenotypes—but not the phenotype being interrogated—is associated with a lower rate of side effects being observed in the interrogated phenotype,

compared with a situation where there is a lack of genetic information for some or all of a drug's targets (OR = 0.91, Fisher's exact $P = 0.0016$; Fig. 2), suggesting that human genetics provides evidence that can be used to de-risk targets.

**Regression modeling.** While a causal link between target genetics and side effects is biologically plausible—genetic and pharmacological perturbation of a protein should lead to similar phenotypes, and this logic is the basis of genetic validation of drug targets[25,26]—there are many potential confounders that could instead be explaining the enrichment we observe. For example, certain kinds of side effects may be correlated with certain drug indications simply because of the patient population treated, and indications will correlate with target genetics due to rational drug discovery and known biology. Indeed, we see that this is a strong correlation within our dataset (Supplementary Figure 1); e.g., drugs with blood-related indications are nearly ten times more likely to have targets with Mendelian genetic syndromes involving blood. Other confounders include variables that have been previously linked to side effects: properties of the protein such as its cross-tissue expression pattern[15], properties of the gene such as evolutionary constraint[37] and corresponding intolerance of population genetic variation[27], or properties of the drug such as modality or delivery route. To explore the relationship of genetics and side effects with these potential confounders, we gathered additional drug and gene information and built a dataset for regression modeling, to predict side effects in each organ system as dependent variables. For independent variables, in addition to the target genetic phenotypes, indications, and modality used for the enrichment analysis, we obtained drug delivery route information, expression breadth information, and population genetic constraint. We encoded all independent and dependent variables as binary values (Supplementary Data 1) and built a logistic regression model for each of 18 side effect classes. Models used all the independent variables described above including the source of

genetic support (GWAS or Mendelian) for the side effect, to predict the presence or absence of the side effect class for each drug. We took two approaches for each side effect model: a multivariate logistic regression model that included all variables, and a machine learning logistic regression model with feature selection. We also ran these models two ways: using the complete dataset and using a dataset that excluded on-indication side effects, as in the previous enrichment analysis.

Strikingly we found consistent, positive coefficients for genetics in models where genetics is significant in predicting drug side effects (Fig. 3; Supplementary Table 3; Supplementary Figures 2–3). This was true even when modeling other predictive variables and excluding potential exaggerated pharmacology (i.e., on-indication side effects). Mendelian and/or GWAS genetics was a significant predictor in the overall analysis for six of the side effect regression models, with four of these (metabolism, heart, endocrine, and reproductive) surpassing a conservative threshold corrected for the number of side effect classes tested (P of coefficient in regression model < 0.05/18). We also found that regression coefficients were similar if expression was included in the model as a matrix of continuous values for each organ instead of as a categorical variable of expression breadth (Supplementary Figure 6 and Supplementary Table 9).

Splitting genetic data into Mendelian and GWAS evidence reduced our power to detect a joint effect on predictive power, but allowed us to ask about their individual contribution. We find that the only side effect classes for which GWAS was independently predictive (heart, mental, and reproductive) were also predictable from Mendelian genetics. However, the raw enrichment results are similar (Supplementary Table 1, Supplementary Figure 4), suggesting that the difference in power is due to a smaller sample size of drugs whose targets have GWAS information compared with Mendelian information. The similarity in effect size between Mendelian and GWAS evidence is

consistent with a previous study of genetic support for drug indications[27].

**Cross validation**. Although the machine learning models perform cross-validation to ensure that the features selected add significant predictive power, there are two potential drawbacks that could lead to inaccurate effect and significance estimates. First, the substantial collinearity between our predictors (e.g., the confounding of indication with genetic phenotypes) may have led to genetics being selected as a predictor even though the causal factor was in fact indication. Second, different drugs in the model often share target genes, resulting in shared genetic information between the training and test sets while the machine learning models are testing new features to add to the model. To address both of these concerns, we implemented a custom leave-one-target-set-out cross-validation procedure to explicitly test the predictive power of genetics (measured as the area under the receiver operating characteristic curve, AUC) in side effect models, when the model has been trained on all other available variables and when the tested genetic data has never been seen in the training set. We use this cross-validation procedure to test the contribution of genetics to off-indication side effect prediction (measured as cross-validation AUC) relative to two baselines: the AUC of a model including no genetic information, and the AUC distribution of models with simulated data with permuted gene–phenotype relationships. We applied these tests to the six side effect models where either GWAS or Mendelian genetics was found to be significant (at P < 0.05). Genetics improved the cross-validation AUC (relative to simulated genetics and relative to no genetics) in five of the six models (Fig. 4; Supplementary Table 4). The absolute increase in AUC is slight but statistically significant, indicating that many of the confounding factors are predictive of side effects, either because they are causal or because they are correlated with genetics. For example, the cross-validation AUC

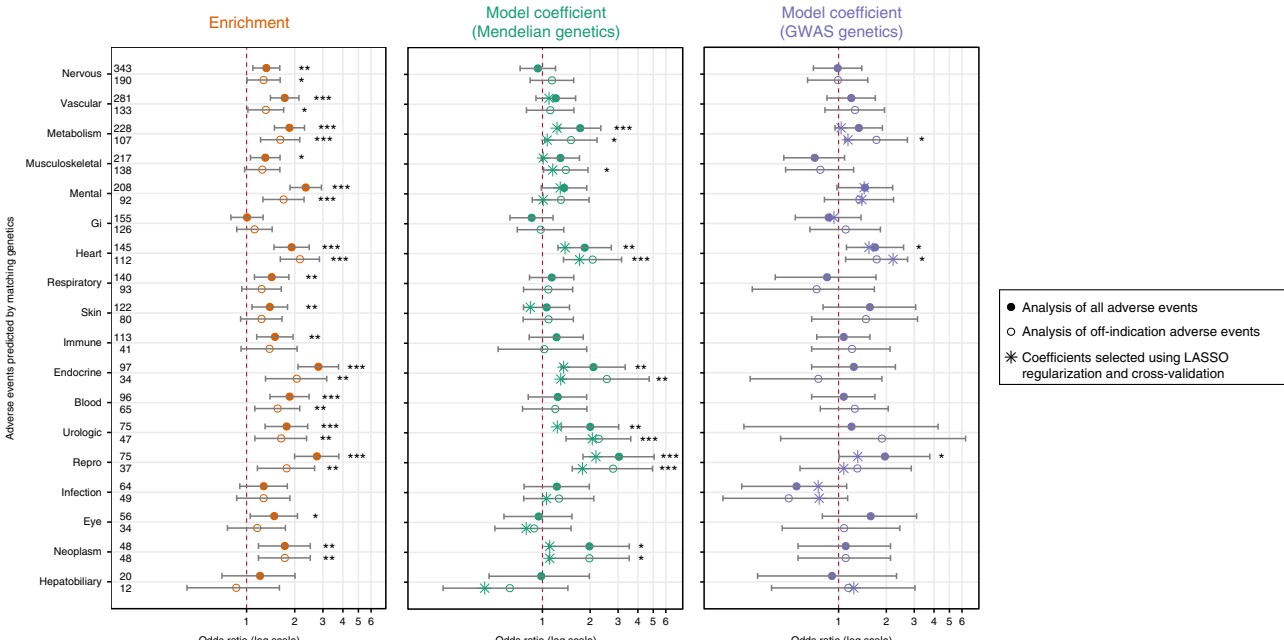

**Fig. 3** Summary of side effect modeling results. Left: results of raw enrichment analysis (source data from Supplementary Table 2). Center: coefficients of the Mendelian genetics predictors in each of the 18 side effect models (source data from Supplementary Table 3). Right: coefficients of the GWAS genetics predictors in each of the 18 side effect models (source data from Supplementary Table 3). Coefficients from the regression models were exponentiated to obtain odds ratios. Along each row, circular points indicate odds ratios from the glm models; error bars are 95% confidence intervals from the glm models; large eight-pointed stars indicate odds ratios when the genetics predictors are selected as predictors in the glmnet models (which are built using feature selection, lasso regularization, and cross-validation). Small asterisks offset from the data indicate P-values of the the Fisher's exact test (left) and P-values of the coefficients from the glm models (center and right) (*P < 0.05; **P < 0.01; ***P < 0.001)

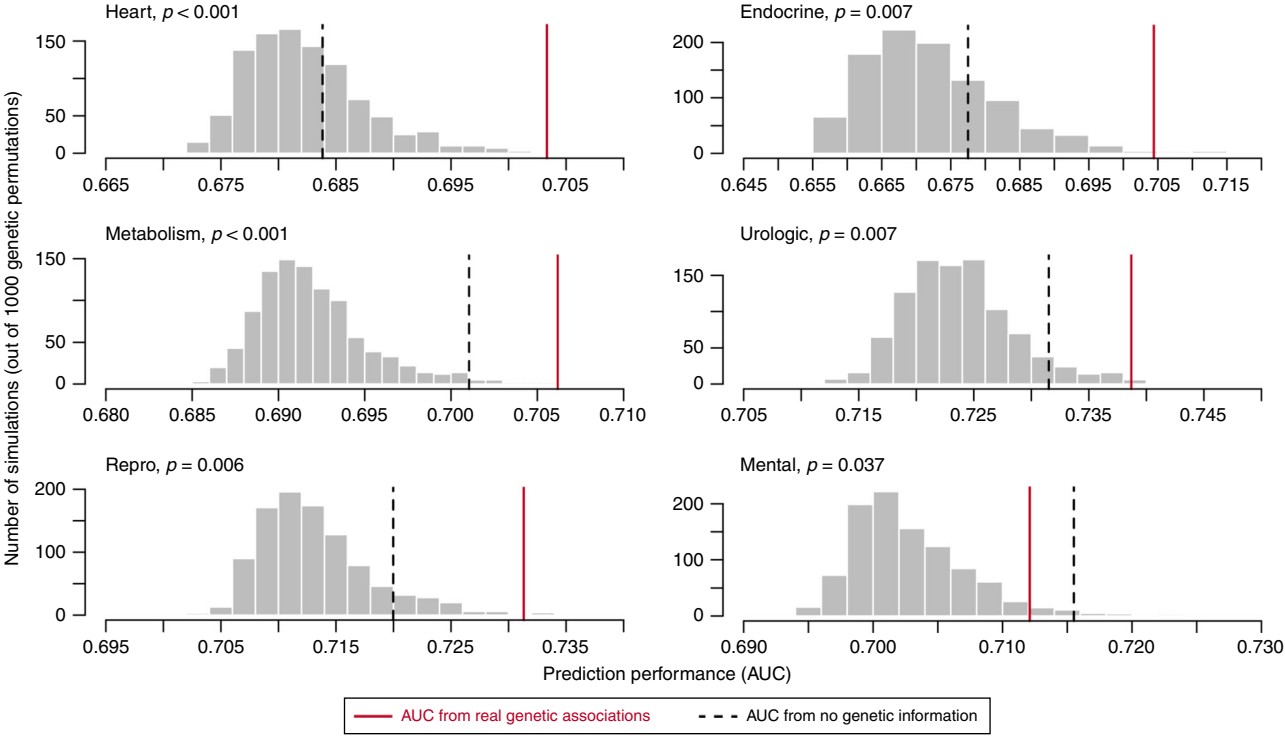

**Fig. 4** Results of cross validation analysis of contribution of genetics to predictive modeling. Gray bars show distribution of leave-one-target-set-out cross-validation AUC values from 1000 permutations of the genetic data; dashed line shows the cross-validation AUC value for a model omitting genetic information entirely; red line shows the cross-validation AUC of the regression model with real genetic data. The P-values are derived from the frequency of permutation runs that exceed the AUC of the true model. Source data are provided in Supplementary Table 4

of the full model predicting off-indication cardiac side effects is 0.70; this drops to 0.68 when omitting genetics and never surpasses 0.70 in 1000 simulations using permuted genetics, evidence that genetics is indeed adding predictive information. The one model of the six tested that did not cross-validate (the model predicting mental side effects) could have failed for several reasons. It could be that in this subset of the data, side effects are more often modulated by off-target effects; the set of drugged targets may be so small and redundant that the effective sample size of independent observations is too small and the analysis breaks down on cross validation; or this class of side effects could be so reliably predicted from other properties in the model that genetics does not contribute additional data, even though genetics may be predictive and correlated with these other drug and target properties.

**Replication using adverse event data**. In order to test whether the relationship between drug target genetics and pharmacological phenotypes replicates in an independent dataset, we used the OFFSIDES database of post-marketing adverse event reports that are inferred to be drug-related[38]. These data are ascertained entirely independently of clinical trial side effects; rather than being reported by clinicians in the context of a clinical trial, they are reported to regulatory authorities by patients and physicians in the context of real-world drug administration. We analyzed a total of 263 non-oncology drugs with at least one human target (Supplementary Data 2), and found that among all 5523 drug-SE combinations, having genetic evidence raised the likelihood of a post-marketing adverse event report from 74.0 to 79.1% (OR = 1.32; 95% CI = 1.14–1.52; Fisher's exact $P$ = 0.00011); this relationship also held true among 4998 off-indication drug-SE combinations, where genetic evidence raised the likelihood of a post-marketing adverse event report from 72.6

to 76.0% (OR = 1.20; 95% CI = 1.03–1.39; Fisher's exact $P$ = 0.016) (Supplementary Table 2).

**Contribution of placebo-associated side effects**. The Cortellis Clinical API attempts to report only drug-related side effects. However, to directly investigate the difference between placebo- and drug-associated side effects, clinical trial data from the aggregate analysis of ClinicalTrials.gov (AACT) was used. Following a processing pipeline in which clinical trials were filtered for confident assignment of drug and placebo trial arms, 229 drugs with at least one human target were used in our analysis. Significantly drug-associated off-indication side effects (Fisher's exact one-tailed $P < 0.05$) were used to compile a dataset of 4275 drug-SE combinations (Supplementary Data 3–5). Among this set, having genetic evidence raised the likelihood of having a drug-associated adverse event from 11.6 to 19.9% (OR = 1.90; 95% CI = 1.56–2.30; Fisher's exact $P = 1.1 \times 10^{-10}$). Without controlling for placebo-associated adverse events, genetic evidence raised the likelihood of an adverse event in the drug treatment arm from 28.6 to 30.6% (OR = 1.61; 95% CI = 1.38–1.88; Fisher's exact $P = 7.3 \times 10^{-10}$). Considering only adverse events reported in the placebo arm, genetic evidence raised the likelihood of an adverse event in the placebo arm from 29.2 to 35.9% (OR = 1.36; 95% CI = 1.16–1.59; Fisher's exact $P = 1.3 \times 10^{-4}$). When these analyses were performed including on-indication drug-SE combinations, results were similar (Supplementary Table 2).

**Discussion**
In this study we show that human genetic studies of drug target proteins can predict not only therapeutic efficacy of drugs, but also can provide evidence about the likelihood of side effects. Although this approach has been proposed and used to make

predictions about individual drug targets[39], it had not been systematically validated. This finding has a number of applications for drug discovery and could ultimately be used to help make safer therapeutics.

First, examining the human genetic associations of potential drug targets early in the drug discovery process could help us more fully understand target biology and select safer targets. Choosing the safest targets at the beginning of drug development should help to make the process more efficient and reduce safety failures that could have been predicted. Previous retrospective analyses of drug development have focused on the relationship between drugs' chemical properties and safety and efficacy[19,20]. Another class of methods attempts to predict the set of off-target proteins that interact with a drug and the resulting side effects[11]. However, such a focus on drug properties and interactions may not help to improve safety if drug developers are not pursuing safe targets. Altogether, our results suggest a practical framework for evaluating a new drug target and for understanding the effects that are seen during clinical development. When making the decision to pursue a novel target, Mendelian syndromes, GWAS associations, and PheWAS databases should be consulted to interrogate the gene(s) encoding the target. If a novel target has been reported as a Mendelian disease gene, all aspects of that disease should be investigated, and researchers should seek to understand whether the genetic disease has aspects unrelated to the intended therapeutic action of the drug. If so, these should be investigated as potential liabilities for the drug; if not, the lack of concerning phenotypes can be used as evidence in favor of target safety. Searching GWAS and PheWAS associations is complicated by the problem of interpreting variants; however, if a variant in a novel drug target gene has an association with the intended therapeutic effect, then this variant can be presumed as functional (in light of other supporting functional genomics evidence) and examined for pleiotropic, non-therapeutic associations. These can then be considered potential target liabilities. The large differences in the nature of genetic and pharmacological perturbation make it difficult to directly translate these phenotypes, their severity, and their clinical relevance. However, targets associated with phenotypes in some of the most critical organ systems (cardiovascular, respiratory, and central nervous system, per international regulatory guidance for safety pharmacology[40]) could prompt critical experimental investigation of these targets' role in those systems.

Second, the genetics of drug targets can continue to inform drug development even after the decision has been made to pursue a target. As drug experiments in preclinical animals and clinical trials in humans proceed, human genetic information can guide researchers to monitor and design assays to test for side effects of most. When side effects do manifest in preclinical species or in clinical trial participants, human genetics of the target can be used to build hypotheses of whether the side effects are target-mediated phenotypes intrinsic to the therapeutic mechanism, or off-target effects that could be engineered out of the drug.

Although the predictive value of genetic target assessment is modest, for some phenotypes the power is similar to the nonclinical-to-clinical power of widely-used animal experiments. An industry-wide analysis of preclinical assays by the IQ Consortium[41] calculated sensitivity, specificity, and likelihood ratios of human translation of positive findings (LR+) and negative findings (iLR−) for phenotypes in various species and organ systems, values we also calculate for target genetic data in Supplementary Table 2. Of the 36 organ system–species combinations analyzed, the majority of positive and negative animal findings lie in the likelihood ratio range that is considered predictive in the sense of increasing the post hoc probability (LR >1),

but not in the sense of being diagnostic (LR ≥10); the same is true for the use of genetic data (Supplementary Table 8). The animal experiments were found to have a sensitivity of 48% and specificity of 84%, while we find that genetic data have a sensitivity of 30% and a specificity of 80%. Despite modest translatability, animal toxicology experiments are nevertheless immensely useful to the drug development process, especially when compared across species and combined with other biological knowledge. We propose that target genetic data should also be incorporated into this wider body of evidence when predicting effects in humans.

Our analysis revealed several examples where side effects would have been suggested as potential issues through examination of target genetics. The first example involves basiliximab, *IL2RA*, and diabetes. New-onset diabetes after transplantation (NODAT) has been noted as a side effect in trials of basiliximab, an IL-2 receptor antibody indicated to prevent transplant rejection; this effect has been confirmed in an observational trial[42] and mouse studies[43]. The *IL2RA* gene is implicated in Type 1 diabetes by GWAS[44], and heterozygous truncating mutations in the *IL2RA* gene cause a Mendelian immunodeficiency syndrome sometimes associated with diabetes[45]. *IL2RA*, encoding the IL-2Rα (CD25) subunit of the trimeric IL-2 receptor, is responsible for high affinity of the receptor for IL-2, and is expressed on activated T lymphocytes including the chronically activated regulatory T (T reg) subset. Imbalance between T reg cells and T effector cells in the pancreatic islet has been implicated in breakdown of self-tolerance and development of type 1 diabetes in a mouse model; IL-2 administration is protective in this model by promoting T reg survival[46].

A second example involves AChE inhibitors, *ACHE*, and bradycardia. Heart rate dysregulation manifesting as bradycardia has been noted in several trials of acetylcholinesterase (AchE) inhibitors for Alzheimer's disease, and this observation has been discussed in pharmacovigilance reports[47,48]; the gene encoding AchE, *ACHE*, has been linked to tachycardia by GWAS[49,50]. Acetylcholine is an important neurotransmitter in the parasympathetic nervous system, which has a cardioinhibitory effect on heartbeat via cardiac M2 muscarinic receptors. AchE inhibition results in the neurotransmitter persisting at the synapse, prolonging the cardioinhibitory effect[51].

The third example involves esreboxetine, *SLC6A2*, and tachycardia. In trials of esreboxetine, a norepinephrine reuptake inhibitor being pursued for fibromyalgia, an increased heart rate has been noted in the treatment arm[52–54]. Esreboxetine acts on the norepinephrine transporter, encoded by the *SLC6A2* gene. Loss-of-function mutation in *SLC6A2* causes a Mendelian syndrome characterized by orthostatic intolerance with tachycardia, resulting from decreased reuptake of norepinephrine and higher plasma concentrations[55,56]. Norepinephrine has a wide range of effects in the sympathetic nervous system through signaling to adrenergic receptors throughout the body, such as increasing heart rate and blood pressure, but reuptake inhibition has been proposed to particularly influence its effect in the heart, because clearance in the narrower synaptic clefts in the heart is far more dependent on reuptake than enzymatic breakdown. This difference is reflected in the fact that *SLC6A2* loss of function affects heart rate disproportionately relative to blood pressure[55].

There are some limitations to our study. The major one is that we are limited by small and biased sample size, both in terms of genetic knowledge and drug trials. Clinical trial side effects are frequently not reported, and our drug trial information is limited to the subset (~12% of trials in the Cortellis Clinical database) that have both outcome data and controlled-vocabulary side effect annotations. We attempt to control for this uncertainty by discarding frequently-noted side effects and by modeling the indication (which will correlate with phenotypes characteristic of

a patient segment). However, individual drug–side effect observations have not been formally adjudicated and may not be treatment related, and conclusions should only be drawn from global patterns. Placebo-controlled analysis of AACT-derived trial data suggests that controlling for placebo-associated side effects improves the predictive power of target genetics by reducing noise due to spurious side effects. The controlled AACT dataset is quite small compared with our Cortellis dataset and as such is not directly comparable. Our sample only includes trials with at least one recorded side effect, which may have biased the analysis toward larger trials. By necessity, our phenotype matching is coarse-grained because of the small sample size and will introduce false positive matches. We anticipate that cleaner, larger sets of clinical trial data would improve our analysis. In addition, there is a paucity of genetic information for some genes either because they are highly constrained or lack common SNPs interrogated in GWAS. Here we do not estimate the direction of genetic effect (e.g., gain- or loss-of function) because this data is not available for the majority of our genetic signals; with better estimates it would be fruitful to compare to the direction of pharmacological modulation. We also lack quantitative information about the magnitude of the genetic and pharmacological effects we consider, and we expect both of these effects to be dose dependent. Different drugs may share the same therapeutic target but have very different side-effect profiles in the clinic due to properties of the drugs, patients, or off-targets; we deal with this by using drugs as the unit of analysis rather than targets, in contrast to the method of Nelson et al.[27]. There is also inherent uncertainty in assigning GWAS signals, the majority of which are noncoding common SNPs, to causal genes; the confidence of this gene–phenotype assignment is limited by the current state of functional genomics data, and adds unavoidable noise to our analysis. We have also made the choice to limit our analysis to targets with known pharmacological action, which means that we end up correlating phenotypes which may, in fact, be driven by off-target interactions with the genetics of known targets. Finally, there are confounding factors that make it impossible to accurately determine the effect of genetics independently of other factors such as indication and delivery route. The raw enrichments are likely overestimates, while the results of the multivariate regression may be over- or underestimates. The results from the cross-validation procedure are likely to be underestimates, since this procedure controls for all other potential covariates before adding genetics to the model; therefore it is these cross-validation results that give us confidence that genetics indeed adds predictive value.

We have demonstrated that the natural experiments afforded by human genetics can be helpful for anticipating the full spectrum of phenotypes elicited by modulating a target pharmacologically—not only the therapeutic effect, but also side effects and adverse events mediated by pleiotropic biological roles of the target. An extension of our work is to understand how the genetics of a drug's off-target proteins also contribute to its side effect profile. Human genetics data are becoming increasingly rich, especially with comprehensive efforts to deep-phenotype complete human knockouts[57–59] and to perform phenome-wide association studies (PheWAS) across electronic medical records to detect pleiotropic effects of genes[39,60]. This analysis underscores the importance of comprehensively characterizing human knockouts and people affected by Mendelian syndromes, because in some cases their biology can help anticipate drug safety issues before they occur, while in other cases their lack of concerning phenotypes can help build conviction that certain proteins are intrinsically safe to drug. Our analysis suggests that this growing body of knowledge will aid in selecting not only more effective targets but also developing safer drugs at lower cost.

## Methods

**Drug databases.** The following pharmacology databases were used: DrugBank[29–32] (v5.0.6, r2017-04-01), STITCH[61] (v5.0, r2016-06-27), Citeline Pharmaprojects (Pharma Intelligence, Informa PLC., d2016-11-22), Cortellis Clinical API (Clarivate Analytics, Inc., d2017-03-31), Cortellis Drug Design API (Clarivate Analytics, Inc., d2017-04-03; filtered for drugs that have been tested in humans, i.e., with highest phase of Phase 0 or higher), and aggregate analysis of ClinicalTrails.gov (AACT) (Clinical Trials Transformation Initiative, d2018-10-18)[62].

To consolidate information about each drug to enter into our model, drugs from DrugBank, Cortellis Drug Design and Citeline Pharmaprojects were merged using an approach combining drug names and aliases/synonyms and CAS numbers (either original active ingredient or its various salt forms). Stereoisomers were also merged (e.g., baclofen and arbaclogen, tretinoin and isotretinoin). This process resulted in unique drug units that aimed to distinguish only active ingredients (but not different formulations or products from different companies; or different cellular expression system for protein-based therapeutics). Drug mixtures, i.e., those with more than one active ingredient, were removed from the analysis for simplicity. To each unique drug unit, intended targets were obtained from the union of DrugBank (targets annotated as having known pharmacological action), Citeline Pharmaprojects annotations, and a recent curated database of therapeutic efficacy targets of a subset of marketed drugs[33]. Target annotations from each database were also separately assessed individually to confirm that the result was not sensitive to the inclusion of any particular database in the union.

Adverse events reported from clinical trials were obtained from Cortellis Clinical API. Of a total of 264,735 trials on record, 92,397 trials had outcomes available. A subset of those trials (31,194) had adverse event terms extracted in computer-readable lists. We limited our analysis to this subset of trials and the corresponding drugs. Trials of combination therapies, oncology indications, and therapeutics that were not small molecules or proteins were also removed. Note that since we could not distinguish between lack of adverse events listed due to complete absence of adverse events or missing data, our dataset only included trials that had at least one adverse event listed as being higher than placebo. We found the frequency distribution of adverse events to be skewed toward a small number of side effects that were very common across different drugs (e.g., headache for 55% of drugs). Because these very common side effects may be less likely to reflect target-mediated effects, and in order to prevent them from overwhelming and biasing the downstream analysis, we excluded 27 of the most common side effects (defined as observed with at least 10% of drugs; Supplementary Table 5), after which 1179 side effect types remained.

For the replication analysis, drug-related adverse events were obtained from OFFSIDES[38]. Drug–phenotype pairs were filtered to control the false discovery rate (FDR) to 5% by implementing a $P$-value threshold of $P < 2.7 \times 10^{-7}$; 184,284 drug–phenotype pairs remained. Drug–phenotype pairs were filtered to retain only those with a reporting ratio (RR) of 2 or greater, a threshold used in the OFFSIDES analysis as a high association score[38]. Common adverse events noted for at least 10% of the drugs were removed. Drugs with oncology indications were excluded. The final validation dataset consisted of phenotypes (genetic and adverse event) for 263 drugs.

Data from AACT were processed to evaluate the contribution of placebo-associated side effects in our analysis. Clinical trials were passed through a filtering pipeline yielding confident assignment of trial arms as either drug treatment or placebo. Only trials assessing a single non-placebo drug were considered. The number of adverse events from the treatment arm(s) of each trial were compared with the placebo arm(s) using Fisher's exact test (one-tailed). Adverse events more common in the treatment arm with $P < 0.05$ were considered drug-related. The final placebo-controlled adverse event dataset contained data on 250 drugs from 488 trials. Adverse events from placebo arms and drug treatment arms were assessed separately as controls.

Drug modality information was obtained from the 'TrialCategories' field of the Cortellis Clinical trial records (either "Biological" or "Small molecule"). Drug administration routes were obtained from the 'DevelopmentStatusSummary' field of the Cortellis Drug Design drug records, and were grouped into four categories: enteral, parenteral, topical, and other (Supplementary Table 6). Drug disease indications were obtained from the Cortellis Clinical trial records and mapped to MedDRA SOC terms as described below.

**Genetics databases.** Genes involved in Mendelian traits were derived from the Human Phenotype Ontology (HPO)[63] (r2017-04-13). In total, 3404 genes were associated with 4390 OMIM syndromes, each of which was described with a list of HPO phenotype terms.

Following the method of Nelson et al.[27], genome-wide association studies (GWAS) associated with each gene were obtained from STOPGAP[34], a database that systematically links published GWAS SNPs results to putative causal genes using a combination of linkage disequilibrium, functional genomics annotations, and variant effect predictions. Only the top-ranked gene (or genes, if tied for top rank) for each GWAS SNP was retained as a gene–phenotype association. Only associations with a genome-wide significant $P < 5 \times 10^{-8}$ were retained, and only associations derived from a publication in the EBI-NHGRI GWAS Catalog were retained. The final GWAS dataset consisted of 4265 genes associated with 431 Medical Subject Headings (MeSH) terms.

**Phenotype mapping**. To allow comparison between all phenotype databases (pharmacology and genetics), we used the Unified Medical Language System (UMLS) Metathesaurus (2015AB release), the MetaMap natural language processing (NLP) tool (MetaMap2016, r2016-01), and the UMLS-Interface software[64] to map phenotypes to the terms found in the Medical Dictionary for Regulatory Activities terminology (MedDRA). All phenotype terms were mapped to the most specific MedDRA term within the UMLS, which was in turn mapped to 26 MedDRA System Organ Class (SOC) terms (Supplementary Table 7). When a single phenotype term was mapped to multiple SOC terms (e.g., 'kidney cancer' is mapped to both 'Neoplasm NOS' and 'Diseases, Urologic'), all SOC terms were retained. All mappings to the SOC terms 'Social circumstances' and 'Surgical and medical procedures' were excluded from analysis. All mappings to 'General disorders and administration site conditions', 'Injury, poisoning and procedural complications', and 'Investigation NOS' were further mapped to other, more organ system related, SOCs using the MedDRA terms one level below the SOCs ('High Level Group Terms').

**Tissue expression and mutational constraint of targets**. Motivated by the observation of Gayvert et al.[15] and others that the general target tissue expression profile is associated with specific toxicities, we used expression breadth annotations from a cross-tissue analysis of the Human Protein Atlas[65]. Following this annotation, genes were put into four categories: all (corresponding to "expressed in all high" or "expressed in all low"), mixed (corresponding to "mixed high" and "mixed low"), enriched (corresponding to "tissue specific", "tissue enriched", or "group enriched"), and none detected. As an alternative to expression breadth, we also incorporated data on tissue-specific expression profiles for each of the drug targets in our analysis to test whether quantitative tissue-specific expression changes the model relative to using a categorical expression breadth predictor. A separate regression analysis was performed with quantitative tissue-specific gene expression values from the Human Protein Atlas[66]. For each drug, the expression values of the drug's targets were converted to the $\log_2$ of transcripts per million (TPM) and then averaged. The 37 tissue-specific expression values were used as covariates in the regression modeling following an identical procedure as described below, with the addition of expression value predictors in place of the expression breadth predictors. Motivated by the previous observation[27] that mutational constraint of targets is associated with drug success, we incorporated the annotation of Lek et al.[67] of Exome Aggregation Consortium variation data to define genes as constrained or unconstrained with a probability of loss-of-function intolerance (pLI) score threshold of 0.9.

**Multivariate regression and feature selection**. To assess the correlation between the genetics of intended targets and the adverse event/side effect profile of a drug, we performed multivariate logistic regressions, both without feature selection (using the "glm" R package) and with feature selection (using the lasso penalized maximum likelihood technique using the "glmnet" R package[68,69]) (v2.0-10). Out of the 21 MedDRA SOC adverse event phenotype groups, we focused only on modeling phenotypes in the 18 groups with a sample size of at least 100 (~5%) drugs eliciting an adverse event in that phenotype group (AE). For each of these 18 phenotype groups, we built a logistic regression model with glm and glmnet using the following predictors: disease indications (20 MedDRA SOCs, excluding "neoplasm"), drug modality ("biological" or "small molecule", with the latter as the baseline), delivery routes from Cortellis Drug Design API, and genetic phenotypes of the drug's targets, encoded as follows: Mendelian genetics for each drug in each side effect model has three possible values: having genetic phenotypes matching the side effect, having genetic phenotypes mismatched to the side effect (there is a syndrome with phenotypes unrelated to the side effect), and having no Mendelian genetic information; and GWAS genetics for each drug in each side effect model has three possible values: having GWAS associations matching the AE, having GWAS associations mismatched to the AE, and having no GWAS hits linked to the target gene(s).

To choose the lambda parameter for the glmnet feature selection procedure, for each model, 10-fold cross-validation was run 100 times, and the final optimal lambda was selected as the average of mean lambda.min (i.e., lambda that resulted in the highest AUC) and the mean lamda.1se (i.e., largest lambda that resulted in an AUC one standard error away from the maximum AUC). The final model coefficients were derived from running 'glmnet' on the whole dataset (1819 drugs) with the optimal lambda.

**Cross-validation procedure**. To assess the predictive power of including genetics in a given side effect model, we performed a custom cross-validation procedure as follows: (1) Each unique target set (combination of target proteins for a given drug) among the list of drugs in the model was enumerated. (2) For each target set, the drugs sharing that target set were held out as tests. (3) The training set was all other drugs, excluding any drug that shared in its target set any of the targets contained in the test set, to avoid any sharing of genetic information between training and test set. (4) Logistic regression (without feature selection) was performed on the training set. (5) Side effect probabilities (the output of the model) were calculated for the drugs in the test set using the coefficients learned on the training set.

A cross-validation receiver operating characteristic (ROC) curve was generated and its area under the curve (AUC) was calculated using the complete set of test predictions. For the side effect models tested, the cross-validation AUC was calculated: (1) With all predictors included in the model; (2) with genetics excluded from the model; and (3) with genetics randomized, such that within the set of drug targets, the assignment of genes to phenotype sets was permuted, maintaining the same fraction of genes with no genetic information and preserving the correlation between genetic phenotypes. This simulation was performed 1000 times to generate a distribution of AUCs.

**URLs**. DrugBank: http://www.drugbank.ca; STITCH: http://stitch.embl.de; Cortellis Clinical API: https://www.cortellislabs.com/page/?api=api-CLI and Drug Design API: https://www.cortellislabs.com/page/?api=api-DD (Note: Information reported in this article is derived from Cortellis for Clinical Trials Intelligence database and Drug Design APIs, which are produced and owned by Clarivate Analytics. Clarivate Analytics will not be liable for any inaccuracy in the information provided in this article or the way in which it is used by any reader in this article.) HPO: http://human-phenotype-ontology.github.io; UMLS and related software: https://www.nlm.nih.gov/research/umls/ (Note: MedDRA® is the international medical terminology developed under the auspices of the International Conference on Harmonisation of Technical Requirements for Registration of Pharmaceuticals for Human Use (ICH). MedDRA® trademark is owned by IFPMA on behalf of ICH.); Citeline Pharmaprojects® | Pharma Intelligence: https://pharmaintelligence.informa.com/; STOPGAP: http://stopgapwebapp.com:3838/SWApp/; OFFSIDES: http://tatonettilab.org/resources/tatonetti-stm.html; AACT: https://aact.ctticlinicaltrials.org/download.

**Reporting summary**. Further information on experimental design is available in the Nature Research Reporting Summary linked to this article.

## Data availability

The datasets analyzed during this study are public, with the exception of two commercial datasets: clinical side effect data, indications, and routes obtained from Cortellis Clinical API and Drug Design API (Clarivate Inc.), and the subset of drug target gene annotations that was obtained from Citeline Pharmaprojects (Informa Plc.), both of which were used under license for the current study, and so are not publicly available. However, complete starting data from which all analyses in the study can be reproduced is provided in Supplementary Data 1 (primary analysis), Supplementary Data 2 (replication analysis using OFFSIDES), and Supplementary Data 3–5 (placebo comparisons using AACT). All other relevant data are available upon request.

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

## Acknowledgements

We thank Cindy Afshari and members of the Translational Systems Biology group at Amgen for valuable discussions.

## Author contributions

P.A.N., D.A.B., P.N., A.M.D. and L.D.W conceived of analyses, interpreted results, and edited the paper. P.A.N., D.A.B. and L.D.W. performed analyses, created figures, and wrote the paper.

## Additional information

**Competing interests:** The authors declare the following competing interest: all authors are current or former employees of Amgen, Inc.

