## [Peer Review File · Nature Communications]

Reviewers' comments:

Reviewer #2 (Remarks to the Author):

This reviewer # 2 previously raised several very important Points of concern.

- 1) The study suffers from a multitude of potentially confounding factors.
- 3) Furthermore, off-target effects are not captured by the proposed model.
- 4) As a result of the above mentioned points the predictive power is further diluted. Thus, even if true, the information provided is not actionable for drug developers. An increase in the probability that side effects will be observed in clinical trials from 19 to 30% will not and should not result in the termination of the respective drug development program, especially if not additional data such as ADR severity or frequency is provided.
- 5) In the discussion section, the authors mention "a number of examples where side effects could have been predicted though examination of target genetics" (p. 4). While for the highlighted three cases, an overlap of genetic associations and drug side effects can be observed, they are not sufficient to conclude that they could have been predicted. If the authors want to make such a bold statement, both sensitivity and specificity of their predictions have to be quantified.
- 6) Including all drugs with "at least one clinical side effect" seems to introduce bias, as trials with larger numbers of participants are at higher risk that one such side effect is observed.

However, the changes in the new version of the manuscript are not enough as to successfully deal with the criticism raised.

Reviewer #5 (Remarks to the Author):

This is a powerful but complex analysis that combines multiple databases in an informed and experienced way. However, that also presents challenges in that the signal could be from multiple sources. A few more controls will increase confidence in this finding.

1. Could the organ system observation be explained by the expression of that gene in that organ system? Of the 38,199 drug-side effect associations what is the enrichment for target expressed (above some cutoff) in that organ class. That should be the null hypotheses. It is not enough to capture expression as enriched, all or mixed as the confounding variable. Please capture expression in each organ class as a variable for the regression analysis.
2. I would also like to see a control where the authors scramble the genetics associations between gene and phenotype.
3. Could you repeat the analysis but take the SEs from the placebo arm in each trial? That could be a good way to control for the indication as opposed to the drug. Also try scrambling the drug SE findings.
4. Line 117 is the Mendelian effect significantly stronger than the combined genetic effect?
5. Line 148 is quite powerful. The implication is that Mendelian disease is supplying all the signal as opposed to GWAS. Perhaps this needs to be emphasized in the abstract. On line 16 please be clear what is genetic evidence (includes GWAS?).
6. Figure 2 please show GWAS data alone as well. The higher OR square brackets are unclear; shouldn't they be from first bar to 3rd bar (rather than from 1.5 to 3rd).

7. It would also be good to see the contribution to the signal from the targets in each of the databases (Drugbank, Pharmaprojects etc). From Drugbank, did you include all the drug metabolism targets such as CYPs as well?

Reviewer #5 (Remarks to the Author):

This is a powerful but complex analysis that combines multiple databases in an informed and experienced way. However, that also presents challenges in that the signal could be from multiple sources. A few more controls will increase confidence in this finding.

1. Could the organ system observation be explained by the expression of that gene in that organ system? Of the 38,199 drug-side effect associations what is the enrichment for target expressed (above some cutoff) in that organ class. That should be the null hypotheses. It is not enough to capture expression as enriched, all or mixed as the confounding variable. Please capture expression in each organ class as a variable for the regression analysis.

We agree that expression breadth could be an oversimplification. To directly investigate this, we integrated tissue-specific target gene expression from the Human Protein Atlas as quantitative variables and reran our regression analysis. Quantitative and tissue-specific gene expression did not significantly change the findings. We have included the results from this analysis in **Figure S5** and **Table S11**, and discussed in the Results section (**Lines 158-159**). This helps us more confidently reject the null hypothesis that the reviewer suggests (that tissue of expression is sufficient to predict phenotypes and genetic information is redundant.)

2. I would also like to see a control where the authors scramble the genetics associations between gene and phenotype.

We scrambled the associations between genes and phenotypes and reran our enrichment analyses. This randomization, like our regression analysis, demonstrates which portion of our enrichment is inflated by collinearity between drugs (rows) and between variables (columns), and which portion comes from genetic information. We find that a significant portion of our enrichment is from genetic information and cannot be explained by other properties of the data structure. We have added this result to **Table S3** and added discussion in the Results section (**Lines 98-106**)

3. Could you repeat the analysis but take the SEs from the placebo arm in each trial? That could be a good way to control for the indication as opposed to the drug. Also try scrambling the drug SE findings.

We agree that placebo-associated side effects are a concern, and that this is an interesting way of controlling for indication and patient segment, and additionally for making our own independent adjudication of whether side effects are drug-related. Unfortunately, quantifying placebo-associated side effects directly is not possible with our main data source (Cortellis). To address this directly, we analyzed data from clinicaltrials.gov through the Aggregate Analysis of Clinicaltrials.gov (AACT). Using this dataset we were able to directly evaluate the difference between treatment arm, placebo arm, and placebo-controlled side effects. Although this data set encompasses a much smaller set of drugs, we reassuringly find that the strongest enrichment is found for placebo-controlled side effects, relative to treatment-arm or placebo-arm side effects, meaning that this increase in enrichment can be interpreted as coming from the drug separate from the properties of the patients. Discussion of these data is included in the results (**Lines 203-215**) and has been added to **Table S3**.

4. Line 117 is the Mendelian effect significantly stronger than the combined genetic effect?

5. Line 148 is quite powerful. The implication is that Mendelian disease is supplying all the signal as opposed to GWAS. Perhaps this needs to be emphasized in the abstract. On line 16 please be clear what is genetic evidence (includes GWAS?).

6. Figure 2 please show GWAS data alone as well. The higher OR square brackets are unclear; shouldn't they be from first bar to 3rd bar (rather than from 1.5 to 3rd).

These three points raise the same fundamental question. We thank the reviewer for probing this issue further, as it raised the fact that we had not analyzed GWAS separately from Mendelian (we had only analyzed Mendelian vs Mendelian + GWAS). This question gave us the opportunity to analyze GWAS separately to answer this question, which we now present separately from an improved and simplified **Figure 2** with corresponding bar charts for comparison in **Figure S4** and data added to **Table S2**. As a result, we saw that the strength of effect was similar for each dataset taken separately. Therefore, we removed any claim about an interpretable difference between the datasets; the difference in significance is likely due to the larger number of drugs targeting proteins with Mendelian information vs GWAS information, which we discuss (**Lines 127-131, 163-165**). We changed the abstract to reflect the top-level aggregate analysis, which addresses the clarification requested by item 5 (**Lines 17-20**).

7. It would also be good to see the contribution to the signal from the targets in each of the databases (Drugbank, Pharmaprojects etc). From Drugbank, did you include all the drug metabolism targets such as CYPs as well?

All target annotations from each source were used, and this can include CYPs if they are annotated as drug targets. We did not include Drugbank “Enzymes” where CYPs are usually annotated, but rather “Targets” with “known pharmacological action” (as noted in the Methods, **Line 360**). We have rerun our enrichment analysis separately using annotations from each individual source of drug target annotations, and seen that the result is similar. Results are displayed in **Table S3** and discussed in the methods (**Lines 362-363**).

Reviewers' comments:

Reviewer #5 (Remarks to the Author):

Thanks for the comprehensive changes in response to the earlier comments.

1 In terms of added results in lines 98-106, the authors should ideally do a 1000+ random permutations and then compute an empirical p-value for the non-randomized observed effect size. This would make this statement more data-driven: " Neither the randomization of genetics nor the randomization of side effects approaches the effect size seen when including true target genetics and drug side effects, confirming the contribution of genetic information to the enrichment, relative to confounding biases (Table S3)."

2 Given that the GWAS data is likely to have higher noise, are you not surprised or concerned that the signal is equally strong for both Mendelian and GWAS?

Reviewer #5 (Remarks to the Author):

1. In terms of added results in lines 98-106, the authors should ideally do a 1000+ random permutations and then compute an empirical p-value for the non-randomized observed effect size. This would make this statement more data-driven: “Neither the randomization of genetics nor the randomization of side effects approaches the effect size seen when including true target genetics and drug side effects, confirming the contribution of genetic information to the enrichment, relative to confounding biases (Table S3).”

We agree that performing the simulation 1,000 times and supplying an empirical P-value would strengthen this statement. We have re-performed the simulation to extend it to 1,000 rather than 100 rounds of permutation. We supplied an additional supplementary **Figure S5** which shows the results of the simulation much more intuitively than the values in Table S3 and in the text. Upon revisiting the text (**Lines 98-107**), we chose to clarify the text to prevent confusion with the Fisher’s P-values and control ORs which are already supplied in Table S3, and instead report the Monte Carlo empirical P ($P < 0.001$) requested by the reviewer, which was the point of the simulation in the first place.

2. Given that the GWAS data is likely to have higher noise, are you not surprised or concerned that the signal is equally strong for both Mendelian and GWAS?

While the difficulty of causal gene assignment is a well-known problem for GWAS, there are also sources of uncertainty for Mendelian knowledge (such as candidate-gene studies). The GWAS and Mendelian literature have different sample sizes and the noise in these analyses may be balanced differently in each set by false positives and false negatives, compounding the difficulty in interpreting differences in effect size and P-value. However, we were reassured by the similarity in effect size between Mendelian and GWAS, because a similar effect size was also reported by Nelson et al. (2015) when comparing the strength of target genetics (Mendelian vs. GWAS) correlation with success of a drug for an indication. Our analysis follows the same biological principles as the Nelson et al. analysis – using genetics to predict pharmacological phenotypes.

We have added this insight to the discussion (**Lines 164-165**) and hope that it clarifies this question in the minds of readers.